# Gender-based disparities and biases in science: An observational study of a virtual conference

**Junhanlu Zhang**[1]*, **Rachel Torchet**[2], **Hanna Julienne**[2,3]*

**1** European Synchrotron Radiation Facility, Grenoble, France, **2** Bioinformatics and Biostatistics Hub, Université Paris Cité, Institut Pasteur, Paris, France, **3** Statistical Genetic Units, Université Paris Cité, Institut Pasteur, Paris, France

\* junhanlu.zhang@gmail.com (JZ); hanna.julienne@pasteur.fr (HJ)

**Data Availability Statement:** In accordance with GDPR regulation, personal data collected in the context of this study are not publicly available. Aggregated data with accompanying scripts to reproduce the manuscript figures have been made

## Abstract

Success in STEM (Science, Technology, Engineering, and Math) remains influenced by race, gender, and socioeconomic status. Here, we focus on the impact of gender on question-asking behavior during the 2021 JOBIM virtual conference (Journées Ouvertes en Biologie et Mathématiques). We gathered quantitative and qualitative data including : demographic information, question asking motivations, live observations and interviews of participants. Quantitative analyses include unprecedented figures such as the fraction of the audience identifying as LGBTQIA+ and an increased attendance of women in virtual conferences. Although parity was reached in the audience, women asked half as many questions as men. This under-representation persisted after accounting for seniority of the asker. Interviews of participants highlighted several barriers to oral expression encountered by women and gender minorities : negative reactions to their speech, discouragement to pursue a career in research, and gender discrimination/sexual harassment. Informed by the study, guidelines for conference organizers have been written. The story behind the making of this study has been highlighted in a Nature Career article.

## Introduction

Gender equity and diversity are key drivers of scientific productivity and innovation. Indeed, women are more likely to include a sex and gender axis into their research [1] as well as challenge long-standing biased practices [2]. Despite ongoing recent efforts, gender disparities and gender biases continue to plague academia and to prevent researchers from reaching their full scientific potential. Especially in STEM (Science, Technology, Engineering, and Math), women encounter various obstacles to career advancement including : hiring discrimination [3], exacerbate skepticism about their contributions [4, 5], and threatening academic climates [6–9]. These biases are observed throughout women's participation in scientific research. For instance, women are disadvantaged in the publication process: women publish less than men [10], are less cited [11], and are less likely to be in the first position among authors who contributed equally [12]. Gender disparities are also noticeable on less externally constrained

publicly available at: https://gitlab.pasteur.fr/hub/gender_at_jobim_2021. To ensure persistence and citability, an additional copy of aggregated data and scripts has been uploaded on Zenodo under the DOI number 10.5281/zenodo.7923694.

**Funding:** The author(s) received no specific funding for this work.

**Competing interests:** The authors have declared that no competing interests exist.

occasions such as question sessions in academic seminars and conferences with women asking significantly less questions than expected [13–18].

Investigators documented this phenomenon in a variety of scientific domains (e.g. genetics [13], neurology [15], hematology [17]) establishing its generality. However, to the extent of our knowledge, no face-to-face interview of conference participants was conducted, leaving the underlying causes unknown. To date, previous study settings were live conferences. The dynamic of gender differences in question-asking behaviors in online conferences could deviate from in-person conferences. For instance, it might be less intimidating to write questions in a chatbox than to stand up and ask a question aloud in front of the audience. This dynamic is important to document since online or hybrid conferences could persist from now on for sanitary, practical, or ecological reasons. The interaction between professional status and gender on question-asking at scientific conferences was only partially investigated so far as it is difficult to know precisely the name and status of askers in in-person conferences. The interplay between these factors could be of interest for several reasons. First, previous reports showed that giving the first question to a student makes the session more welcoming to women [13]. Second, in non-academic settings (United States Senate), a study documented that powerful women speak briefly by fear of backlash while powerful men do speak lengthily [19]. The online setting allows for a finer identification of question askers and to retrieve both gender and professional status. The observation of an online conference might hence shed more light on this power dynamic in STEM.

Here, we aim to document the impact of a virtual setting, gender, and professional status on question-asking behavior during scientific conferences and to provide hints to mitigate the underrepresentation of women and gender minorities amongst question askers. We comprehensively observed the JOBIM (Journées Ouvertes en Biologie et Mathématiques) 2021 conference (S1 Fig). This conference is a convivial, medium scale event ($\tilde{3}00$ to 600 participants) that has been gathering annually the French bioinformatics community for the last 20 years. The JOBIM 2021 conference was for the second time held online due to the COVID-19 pandemic (The JOBIM conference was held online for the first time in 2020). The virtual setting offers us a unique opportunity to understand the differences between in-person and online question-asking from a gender perspective. During the conference registration, we collected demographic data and included the possibility to disclose sexual orientation as an important demographic characteristic often left out in previous studies [20]. We attempt to step out of the historical or typical binary vision of gender by giving the attendees the opportunity to report their self-identified genders with options other than "man" and "woman". We also investigated the reasons behind women and gender minorities under-representation in question askers by conducting in-depth interviews of participants and a post-conference survey.

We believe that to successfully address women and gender minorities' (by gender minority we refer to individuals identifying outside of the binary gender categories that are male or female) under-representation at scientific conferences, we need a clear picture of its causes that can only be derived by studying their overall experience. We therefore adopted an evidence-based and mix-method approach to delve into gender-based disparities in science. Being at the interface between hard and soft sciences, this study integrates qualitative and quantitative research methodologies. Considering the complex and intangible nature of gender biases, we believe that bringing in-depth qualitative analysis into the picture is unarguably important, as illustrated by previous work [21]. In this study, we provide an overview of the evolution of the JOBIM conferences demographics across two decades and an analysis of question-asking behavior. We then contextualize these results with the perception of JOBIM

participant collected through a post conference survey and interviews. The story behind the making of this study has been highlighted in a Nature Career article [22].

## Materials and methods

### Demographic analysis of previous conferences

Anonymous data on previous editions on attendees, speakers and keynote speakers was provided by the SFBI. When asked at registration, gender was inferred from civility. For editions where civility was not recorded, gender was inferred by the SFBI from the first name of the participants before the anonymous data was handed to us. Gender most commonly associated with the first name was looked up in a public database of French first name (https://www.data.gouv.fr/en/datasets/liste-de-prenoms/). The quality of such a procedure was assessed using data from the 2021 editions by comparing the gender proportion assessed by this method and the one measured by self-identified gender (S1 Table). While gender could not be identified from the first name in 12.5% of the cases, the inferred gender proportion was still accurate and equivalent to the proportion computed with self-identified gender (two-sample proportion test p-val 0.94, self-identified proportion : 0.508, first name proportion : 0.503).

To assess the proportion of women in committees, the names of committee members were retrieved from the website of past and current editions of JOBIM (from 2015 to 2021). Each committee member was assigned a gender from their first name by looking up the corresponding gender in the first name database. When gender could not be assigned from the first name alone (notably for rare or foreign first name for instance), the gender was assigned by looking up a photograph of the committee member. The gender assigned by this procedure is the gender as perceived by society and might differ from the gender identity.

### Registration and post-conference surveys

To understand the factors that potentially contribute to attendees' question-asking behavior during the conference, we invited all attendees' to fill out relevant questions through online surveys. Registration survey is a mandatory step when the attendees sign up for the JOBIM conference. In the survey, apart from asking regular questions on the attendees' demographic profiles including age and country of residence, we also gave the opportunity for the respondents to share personal information on their self-identified gender, preferred pronouns and whether or not they identify with the LGBTQIA+ community. We made sure to acknowledge all survey respondents that sharing their personal information is not a mandatory part of the registration procedure, which means not answering those questions will have no impact on their registration to the conference.

A survey was also sent after the JOBIM 2021 conference. The purpose of the post-conference survey is to collect data related to the attendees' question-asking behavior during the conference as well as their previous experiences with and general opinions on academic conferences in STEM. Therefore, we divided the post-conference survey into three sections: contact information, question-asking during the JOBIM 2021 conference, and general experiences at conferences (Fig 3). On the 695 participants at JOBIM 2021, 525 consented to the use of their data for this study and 152 answered the post-survey.

### Observation of the virtual conference

During the JOBIM 2021 conference, an observational study was conducted to collect data that is relevant to attendees' question-asking behaviors. Given the fact that the conference took place virtually on the videoconferencing platform Zoom, the research team gathered

observational data through : 1) an observation forms filled out by a team of observers and 2) attendance reports generated from Zoom.

The observation team consisted of 8 observers (6 women, 2 men), including researchers from the research team as well as conference attendees who volunteered to support the observational study. Prior to the conference, an online version of the observation form was developed and tested to assist the recording of observational data. During the conference, observers followed a set of guidelines (S1 File) to observe the planned sessions for observation and to fill out the observation form accordingly. The observation form was a key tool to collect data on question-asking behavior, for example, the number of questions asked during each session, type of the questions, and gender of the question askers (gender of the askers was identified by the observers). To minimize the human errors that could potentially occur during observation, each session was observed by at least 2 observers. By the end of each session, observers submitted the observation forms online. Zoom attendance and Q&A report were exported for each session. The gender of question askers was identified using the following procedure: 1) Use self-identifying gender from the registration survey when available, 2) Identified from first name, 3) in last resort, we queried the name of the asker through Google search engine and identified gender from a portrait. Q&A report were manually curated to remove comments containing no question ("thanks", "clap-clap", . . .). In rare occurrences, one question was spread over several postings. In this case the question was counted as 1 item.

## Statistical analysis

Statistical analysis were conducted using R/3.6.3. Underrepresentation of women and gender minorities in question askers compared to attendees was tested using the exact Fisher test. Comparisons of postsurvey answers between genders were performed though a $\chi^2$-contingency table test.

For the Poisson regression of the rate of question asking by demographics, we computed the number of questions asked by each attendee by counting their corresponding entries in Zoom's "Q_n_A" reports and the total time spend in conference by summing up the "Time in Session (minutes)" column in the Zoom's "Attendee" reports. The rate of question asking was the number of questions asked over the total time spent following the conference. The model retained by the AIC criteria (S2 Table) is

$$
\begin{aligned}
ln(Y) = \beta_0 \quad &+ \beta_1 \times (Gender = Male) \\
&+ \beta_2 \times (Age \geq 35) \\
&+ \beta_3 \times (ProfessionalStatus = permanent) \\
&+ \varepsilon
\end{aligned}
\tag{1}
$$

where $Y$ is rate of question asking.

## Semi-structured and in-depth interviews

Qualitative data was collected through 7 semi-structured and in-depth interviews 2–3 weeks after the JOBIM 2021 conference. he number of interviews we were able to conduct is relatively low (7) and our qualitative observations might not generalize to the complete academic population. Yet, these interviews provide valuable examples of career trajectories impacted by gender and by the LGBTQIA+ status. This contextualization can help generate hypothesis on why women and gender minorities ask less questions than men. To reflect the diversity of the population studied and to respect the intersectional basis of this study, we created a

customized interviewee selection procedure based on the principle of quota sampling. Attendees' age/seniority, gender, nationality, membership of the LGBTQIA+ community as well as activeness (activeness specifically refers to number of questions asked during the conference) during the conference were taken into consideration. Attendees who responded that they were interested in or may be interested in the post-conference interview during the registration process were categorised into the following subgroups: "junior male attendees" who self-identify as male and are under 35 years old, "junior female attendees" who self-identify as female and are under 35 years old, "senior male attendees" who self-identify as male and are above 44 years old, "senior female attendees" who self-identify as female and are above 44 years old, "foreign female attendees" who self-identify as female and are foreign (non-French) nationals; "LGBTQIA+ attendees" who self-reported as members of the LGBTQIA+ community, and "serial question askers" who asked 4 and more than 4 questions during the conference. A random interviewee was selected from each of the subgroups and contacted for the interview. It is worth mentioning that, due to the lack of response from the contacted interviewee candidates, the interviewee of the subgroup "senior male attendees" was identified among the younger age group "between 35 and 44 years old". All interviewees agreed to attend an one-to-one online interview session on the voluntary basis. Prior to the interview, each interviewee was asked to carefully read and sign the consent form (S2 File) which provides description of the project, detailed information about the interview as well as data privacy and protection policies.

Following the principles of semi-structured interview, we developed an interview guide (S3 File) with the purpose of ensuring the organisation and flexibility of the interviews. The interview guide served as a checklist of topics to be covered and provided a set of interview questions. Although the questions were designed beforehand to guide the conversation, the interviewer prioritized open-ended questions and encouraged two-way communication during the interview in order to collect qualitative and in-depth information. All the interviews were recorded via the video-conference platform (All the interviews were conducted on ZOOM) and the recordings were later used to produced interview transcripts. The interviews were between 33 and 58 minutes.

## Qualitative data analysis

To analyse data collected from in-depth interviews, we adopted qualitative coding as the key method to transform data into findings with the support of one of the popular CAQDAS (Computer-assisted Qualitative Data analysis Software), MAXQDA. The process of analysing qualitative data in this study combined inductive and deductive approaches. On the one hand, an initial codebook S3 Table) was developed prior to coding, based on research questions and interview guidelines. Based on the codebook, researchers were able to apply predefined codes to the interview data. On the other hand, line-by-line coding was performed on all interview transcripts. Associating with the constructivist grounded theory, this approach made sure that close attention was paid to all collected data and that researchers had the opportunity to extract as much information as possible from the data. To generate conclusions from the coded data, the initial codes—including parent codes and child codes—were grouped into concepts of higher order named categories (S4 Table). As a result, the categories were developed into themes which represent the significant findings of the study. Developing themes involve a necessary process of comparing identified codes regarding their importance, particularly based on the following two measures: 1) the frequency of a code occurring within one specific information source, and 2) the frequency of the code occurring throughout all information sources.

### Ethic statement

Participation in this study was independent of JOBIM 2021 conference attendance. All of the people who took part in the study gave explicit and written consent to the processing of their data for the research project, in particular to the processing of sensitive data within the meaning of article 9 of the GDPR (data concerning sex life and sexual orientation), in accordance with the requirements of the GDPR. The personal and sensitive data collected in the context of this study was deleted once the objectives of the study were fulfilled. The detailed data policy was also available on the project page (https://research.pasteur.fr/en/project/jobim-2021-pilot-project-gender-speaking-differences-in-academia/). The data processing has been entered in the records of Institut Pasteur processing activities with the assistance of the Institut Pasteur data protection officer. This study was declared exempt of ethical concerns by the Institut Pasteur IRB (IRB00006966).

## Results

### Demographic of current and past JOBIM editions

JOBIM targets young professionals in bioinformatics and is known as a convivial event where attendees participate in a series of scientific talks and social events. A striking feature of JOBIM 2021 conference confirms its popularity among young attendees: based on the registration form, a majority of attendees declared they are under 35 years of age (62%, Fig 1A). Parity between female and male attendees has been reached for age categories under 45 (52% of women) but not among attendees who belong to the older age category (38%, p-val = 0.047, Fig 1A). Concerning gender, the vast majority of attendees identified themselves within the binary gender categories (i.e. 'Male' and 'Female'). However, 13 attendees preferred other categories (1 agender, 4 Non-binaries, 8 prefer not to say). A substantial fraction (9.5%) of participants self-identified as members of the LGBTQIA+ (Fig 1B). Answering Yes to the question 'Are you a member of the LGBTQIA+ community?' was more likely in young attendees (Fig 1B). Women attendees were also more likely to identify as members of the LGBTQIA+ community (13% in women versus 5% in men), which is consistent with a previous report stating that lesbian and bisexual women are as likely to graduate and work in STEM (compared to straight women), whereas gay men tend to drop out more than straight men [23].

We equally retrieved the number of participants and their gender from past editions (inferred from first name, Materials and methods). The proportion of women in the audience has increased steadily at a slow pace from the first edition in 2000 to 2019 (Fig 1C). Yet, in all these editions, women were significantly underrepresented in participants compared to the general population. In 2020, JOBIM became virtual due to the COVID-19 pandemics. Simultaneously, the proportion of women increased and was no longer significantly below 50% for the 2021 edition. To disentangle the slow increase of women proportion in the field with the effect of virtual conferences, we compared the proportion of women in the 2 last virtual editions with the 2 last in-person editions. The women proportion difference between virtual editions and recent in-person editions was highly significant (in-person proportion: 40%, virtual proportion : 48%, p-value = $3.4 \times 10^{-4}$) suggesting that virtual conferences might increase women attendance. The proportion of women in contributed speakers (who submitted an abstract to the conference and were selected for giving an oral presentation.) varies with the year and is not always significantly below 0.5 (Fig 1D). Yet, amongst the 13 JOBIM editions with available data, 6 significantly lacked women contributed speakers. We retrieved the number of posters presented for this edition and we did not observe a gender imbalance (S5 Table). An

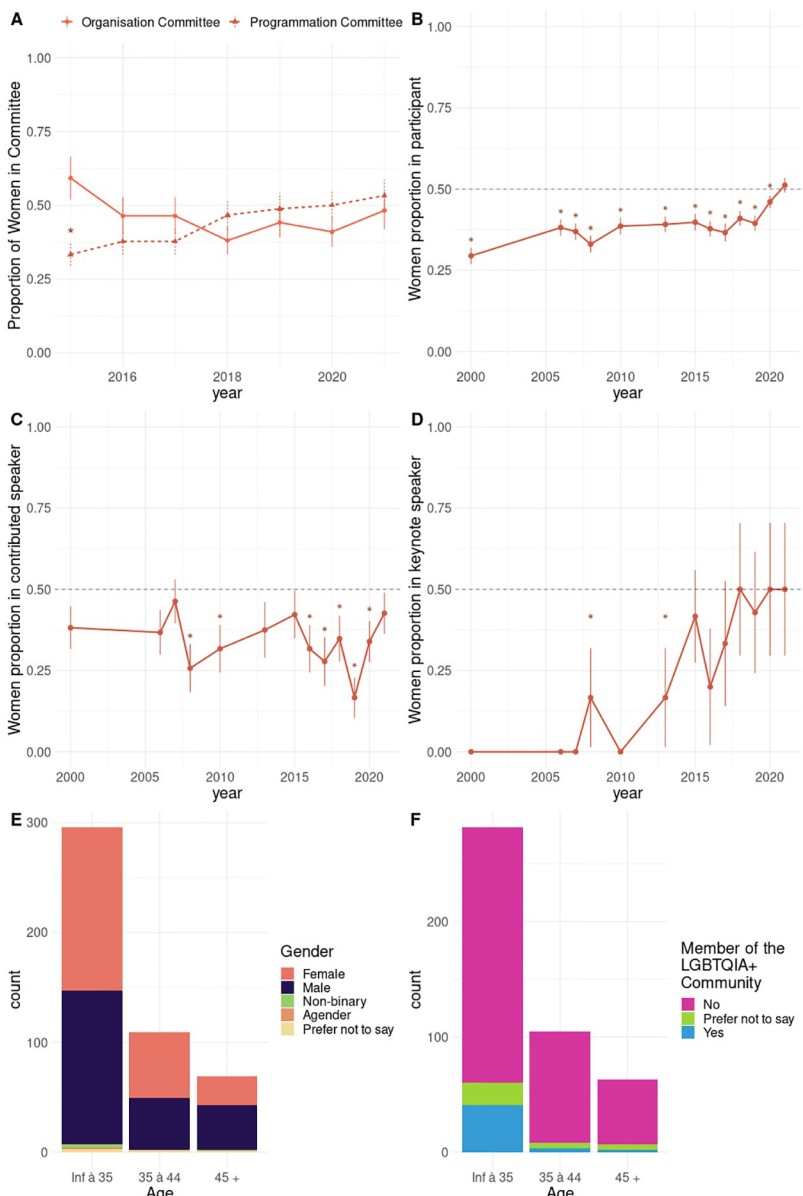

**Fig 1. Demographics of JOBIM conferences in current and past editions.** A) Count bar plot of the number of participants at the JOBIM 2021 conference by age category. Colors represent the count of each gender. B) Count bar plot of the number of participants at the JOBIM 2021 conference by age category. Colors represent the attendee answer to the question: "Do you identify as a member of the LGBTQIA+ community?". (C-F) The proportion of women with respect to JOBIM edition year in : C) participants, D) contributed speakers, E) organization and program committees (line types represent the type of committee), F) keynote speakers. In plot C, D, E, and F vertical bars represent the standard error on the proportion estimate. Stars indicate a significant deviation from parity.

improving trend is noticeable in the composition of committees and keynote speakers. The proportion of women in the program committee has increased in 2018 and remains stable since (Fig 1E). No "manel" in keynotes was observed since 2010 (Fig 1F). Interestingly, the parity in the program committee and keynote speakers seems to be simultaneous. These tendencies are in line with previous reports [24] showing that the presence of women in committee efficiently diminished the chance of invited speakers being all men.

### Written question asking

Throughout the JOBIM 2021 conference, 192 questions where asked through the chatbox during a variety of scientific sessions: 57 questions asked by women, 115 by men, and 20 by anonymous or group attendees. No question was asked by a person from a gender minority (agender, nonbinary, or transgender). Since gender parity was reached among the conference attendees, women were significantly underrepresented in question askers (Exact fisher test p-value : 3.1e-05, Fig 2A and 2B). We noticed a tendency for a stronger underrepresentation in 'Mini-symposia', which are series of technical talks featuring invited speakers, and less obvious in the contributed talks, which usually feature more junior speakers (Fig 2A). This imbalance is further reinforced when weighting the questions by their length (Fig 2B). Similarly, previous reports stated that in live conferences men ask more oral questions and slightly longer ones [13, 14]. The proportion of women in sessions did not deviate strongly from parity in attendees (range:43% to 57%) while being under 50% in 14 sessions out of 23 for questions asked (range: 0 to 66%). The number of questions asked by members of the LGBTQIA+ community was slightly lower than expected based on the registration data, yet this tendency was not significant (Exact Fisher test odds ratio : 0.4, p-value: 0.067).

We leveraged the Zoom exports to compute the number of questions asked by participants throughout the JOBIM 2021 conference (Materials and methods). The vast majority of attendees did not ask questions during the conference (87%). The overrepresentation of men is more pronounced in attendees asking several questions throughout the conference (Fig 2C).

To understand the interplay between professional status, age, gender and being members of the LGBTQIA+ community, we modelled the rate of question asking as a function of these factors using a Poisson regression (Materials and methods). We compared several models including different sets of variables. According to the AIC criteria (Akaike information criterion), the two best models comprised gender, age, and status and the interaction between status and gender as predictors (S2 Table). However, the interaction between status or age and gender was not significant on its own. Hence we focused on the model including age, status and gender. When accounting for age and professional status, the effect of gender remained significant, with men asking twice as many questions as other genders (Table 1). Even though

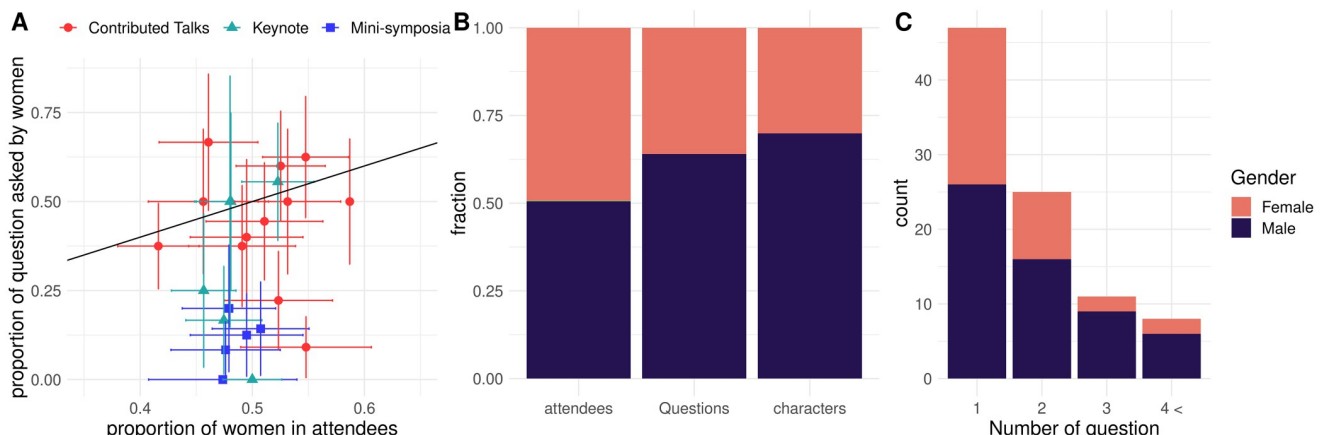

**Fig 2. Written question asking during the JOBIM 2021 conference and gender.** A) Proportion of questions asked by women with respect to the proportion of women in attendees for each session. The color and shape of point represent the session type. Horizontal and vertical bars represent the standard error on the proportion in askers and attendees respectively. The black diagonal line represent the expected proportion in askers accounting for the proportion in attendees. B) Barplot of the proportion of women in: 1) attendees, 2) questions asked 3) total number of characters written. C) Count of askers by the total number of question they asked throughout the conference. Colors in barplot represents the gender of the asker.

**Table 1. Model of the rate of question asking by hour as a function of gender, age and professional status.** A Poisson Regression of the rate of question asking as a function of gender, age and professional status was fitted to the data. The exponent of the intercept corresponds to the question-asking rate of an attendee who is a woman or a gender minority, under 35 and have a short-term contract. The exponents of other coefficients give the multiplicative factor to apply to the rate when the condition is verified.

| | $\beta$ | $e^{\beta}$ | p-value |
|---|---|---|---|
| Intercept | -4.34 | $1.13 \times 10^{-2}$ | - |
| Gender = Male | 0.7 | 2.01 | $1.34 \times 10^{-4}$ |
| Age > 35 | 0.71 | 2.02 | $1.05 \times 10^{-2}$ |
| Permanent position | 0.823 | 2.28 | $5.5 \times 10^{-3}$ |

JOBIM is considered as student-friendly conference, the effect of age and professional status weighed heavily on the rate of questions asked with senior academics asking 4.6 times more questions than junior academics. When modulating this effect by gender, senior women and gender minorities would ask 2.3 more questions than junior men, while senior men would ask 9.3 more questions than a junior women or gender minorities. Note that while we did not find an interaction between gender and seniority using Poisson regression, the multiplicative nature of the model implies that seniority for men results in a larger increase of the rate of question asking than in other gender categories (S2 Fig).

## Questions asked or read out loud

The results could differ from the preceding section due to a few questions asked orally by the attendees, the questions asked by the chairperson, and the selection effect of the chairperson when reading questions out loud from the chatbox. We counted 257 questions by observing all JOBIM 66 talks. Only 78 questions (30.3%) where asked by women despite a proportion of attendees close to parity (woman proportion : 50.2%). The difference between the two proportions is highly significant (proportion test p-value: 4.44e-10). We did not observe a specific effect of the gender of speakers or chairpersons, contrarily to previous studies (although the sign of the effect was not consistent from one study to another) [13, 14]. The number of chairwomen and chairmen was equivalent in the JOBIM 2021 conference (36 and 37 respectively) and the number of question they asked followed the global trend with chairwomen asking less questions (16 and 31 respectively, OR = 0.53).

The total number of questions asked at the end of a talk depends on the type of session, with keynotes receiving twice as many questions (p-value: 0.01e-2) as mini-symposia talks and contributed talks (S3A Fig). While observing a trend consistent with a previous report [14], we did not find a significant positive effect of the total number of questions asked during the question session on women and gender minority representation (S3B Fig).

## Post survey results

Participants shared their thoughts on what factors influenced their question-asking behaviors and on their experiences at scientific conferences in general. Of the 695 registered participants, 144 shared their feedbacks with us through a post-conference survey (Material and methods).

Overall, factors encouraging, discouraging or motivating question asking were considered to be the same by all genders (Fig 3A–3C), which suggests that a welcoming environment would be the same for everyone independently of gender. Top impacting factors are similar research interests between the asker and the speaker, clarity of the talk, and confidence. Note that our results differ from [14] which showed that factor hindering question-

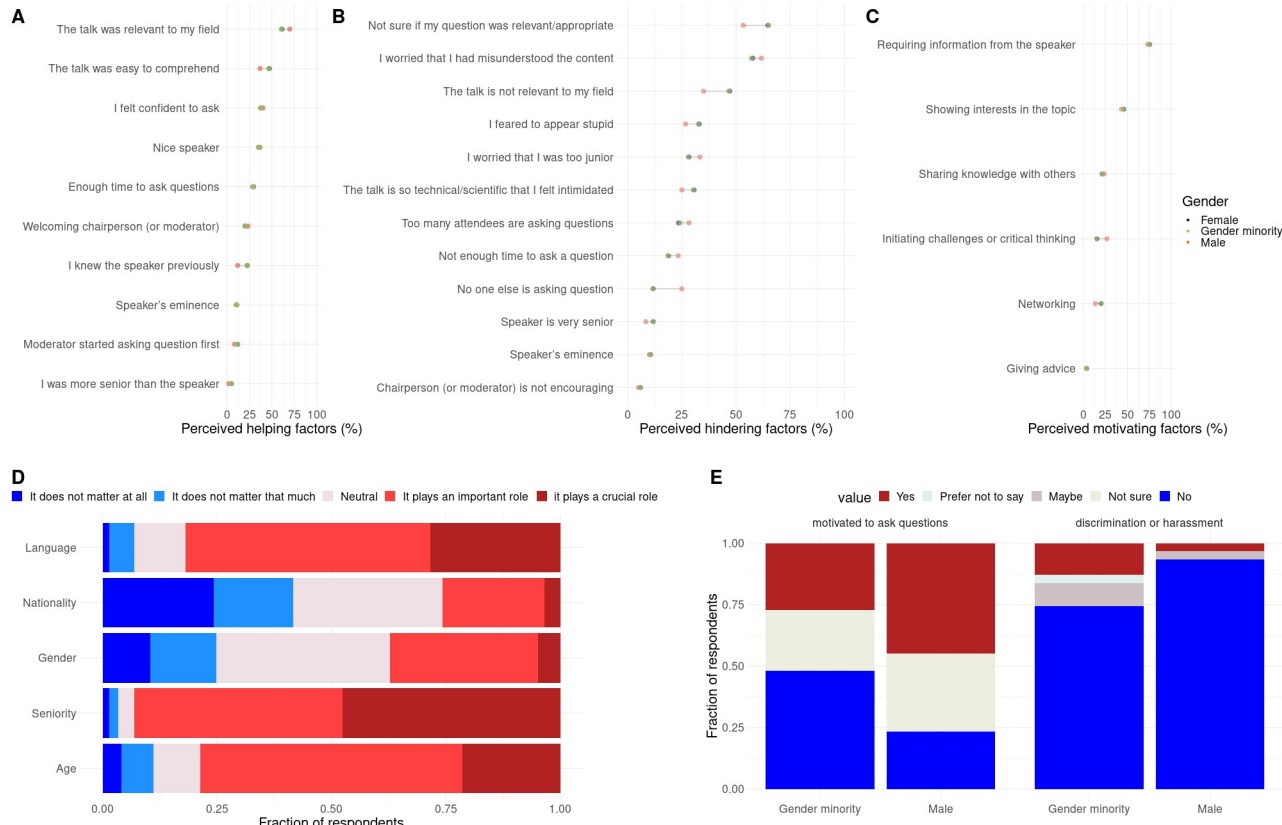

**Fig 3. Post survey results.** A) fraction of respondent answering these factors encourage them to ask questions, B) fraction of respondent answering these factors discourage them to ask questions, C) fraction of respondent answering these factors motivate them to ask questions, D) Likert scale on the perceived importance of several factors on question asking E) conference experience items significantly differ among genders.

asking where rated as more important by women. This might be due to the survey design : [14] used Likert scales (where respondents are asked how much a factor matters using a scale from one to five) whereas we asked respondents to say if a factor mattered or not in a binary way.

Gender did not impact the perception of what demographic factor was important regarding question-asking Fig 3D. The effects of age and seniority were clearly perceived by respondents with respectively 80% and 94% of respondents selecting "it plays an important role" or "it plays a crucial role". In contrast, 38% of respondents thought that gender was an important or a crucial factor for question asking (Fig 3D). While none of the respondents thought that women asked more questions than men, a large fraction 50% was "not sure about" which gender ask more questions.

While for most survey items all genders provided similar answers, there were a few notable exceptions (Fig 3E). Women and gender minorities declared to be less motivated to ask questions at conferences in general (Chi-squared test p-value = 6.4e10–3). Women and gender minorities were more likely to declare that they have or might have experienced discrimination or harassment during conferences (answering "Maybe" or "Yes", Exact Fisher test p-value = 4.0e10–3, OR = 4.6). This observation is coherent with previous reports stating that women experience is overall harsher in STEM than men experience [25–27]. This strongly emphasizes the need for guidelines and for a reporting system during conferences.

### In-depth interviews : Gender-based differences experienced at scientific conferences

To contextualize quantitative findings we conducted in depth interviews on a representative set of participants. The number of interviews we were able to conduct is relatively low (7) and our qualitative observations might not generalize to the complete academic population. Yet, these interviews provide valuable examples on how gender and LGBTQIA+ status can impact career trajectories which could lead to a lesser comfort in question asking.

We selected an interviewee for each of the 7 classes identified during the quantitative analysis (Material and methods) : "junior male attendees", "junior female attendees", "senior male attendees", "senior female attendees", "foreign female attendees", "LGBTQIA+ attendees", and "serial question askers". By analyzing the transcripts of 7 interviews, we developed a codebook collecting recurring themes in interviews (Material and methods, S4 Fig, S3 Table).

Findings indicate 4 main recurring topics: 1) women and gender minority attendees report negative experiences based on sexual orientation, gender identity and expression in professional contexts; 2) there is a significant correlation among gender identity, confidence and career advancement; 3) women and gender minorities are more proactive to challenge gender inequalities in the workplace; and 4) women report experiencing gender-based discrimination and harassment during scientific activities, including academic conferences.

### Women and gender minority attendees report negative experiences based on sexual orientations, gender identities, and expressions

Negative experiences reported by women and gender minorities during interviews can be grouped into two types—"discouragement" and "professional abilities being undermined" due to their sexual orientation, gender identity, and expression. Three female interviewees stated that they felt discouraged to ask questions in conferences or in the workplace by experiencing or witnessing negative reaction to women's speeches.

"When she (female team leader) was open to ask questions or even to present, she was very...they (male team leaders) were not nice with her." (I4—Senior foreign female attendee)

Another female interviewee expressed that she was discouraged to become a scientific researcher already in the early stage of career planning.

"My parents were saying that I should be a teacher instead of a researcher, (this way) I would have more holidays for my children. At the time I thought... wow I don't want it. I don't want to my professional to depend on holidays and kids. I want to be a researcher. I must say that... my brother who is younger than me, they (the parents) were not saying that to him." (I3—Senior female attendee)

Regarding women and gender minority professional abilities being undermined, 2 female and gender minority interviewees indicated that they were either labelled or alienated based on their sexual orientation, gender identity and expression.

"I was often known as the 'gender nerd' in my field. I am not sure I want to be that, because sometimes you are perceived to be just that." (I5—Junior female attendee)

### Correlation between gender identity, confidence and career advancement

Qualitative analysis of the interview transcripts suggests a correlation between gender identity, confidence and career advancement. To be specific, 3 female and gender minority interviewees pointed out that their gender identity impacted their level of confidence.

"...That's also possible (that men speakers are more confident). Maybe they just don't often have negative reactions to their questions." (I2—LGBTQIA+ attendee)

In turn, 3 female and gender minority interviewees stated that confidence is a key factor which influences their career advancement.

"Interviewer: what do you consider as the biggest barrier in your career path?.

Interviewee:I think (it's) self-confidence. Self-confidence is really impacted by how society reflects me as a woman and the ways genders are expressed." (I5—Junior female attendee)

Furthermore, the findings also show that women and gender minority might improve their level of confidence through self-acceptance. As an example, a gender minority interviewee reported that the reason behind gaining confidence is rather "internal".

"(I feel more confident in speaking up). . .Because I do feel more confident as myself, as my own person. That just makes it easier. (I2—LGBTQIA+ attendee)"

In comparison, 1 male interviewee suggested that his level of confidence has been improved through professional experience.

"Now that I am more senior (in research), I am asking more questions and I am less afraid of asking questions. (It was) something I didn't do when I was younger. (I1—Senior male attendee)"

These observations correspond in part with studies reporting that women tend to have a lessened self-esteem compared to men, especially during early adulthood, and start recovering partially during their forties [28, 29]. Our observation suggests that the origin of this recovery could be internal rather than external. Confirming these hypotheses would need further investigations.

## Women and gender minorities are more proactive to challenge gender inequalities in the workplace

During the analysis of interview data, interviewees displayed various attitudes towards the topic of gender inequality, including conscious, curious, supportive, oblivious, and "active" (i.e. interviewees taking actions or being eager to make a change). Findings show that all inter-viewees were conscious about the subject of gender inequality either in the workplace or in society. However, female and gender minority interviewees presented a more proactive atti-tude to tackle gender inequality or more willingness to make a change.

"As a woman, I am happy to be in this field because I can help and to improve (gender inequality). (I4—Senior foreign female attendee)"

Male interviewees stated that they were oblivious to potential issues related to gender inequality or they did not feel concerned by the issues.

". . .But maybe I heard it and I don't remember, because I am a male and it doesn't impact me. (I1—Senior male attendee)"

These findings are consistent with previous reports stating that men are more often unaware of these inequalities than women and gender minorities and are more skeptical about empirical studies on inequalities [30, 31].

## Women are subjected to gender-based discrimination and sexual harassment

Gender-based discrimination and harassment in the workplace (i.e. "gender-based behaviors, policies, and actions that adversely affect work by leading to disparate treatment or creation of an intimidating environment", and Sexual harassment covers "a spectrum from generalized sexist remarks and behaviors to coercive sexual advances and from unconscious patronization and subtle innuendo to blatant sexual threats" [32], including academic conferences, were one of the most frequently discussed phenomena during the interviews. The subject of discussion was either brought up by the interviewees voluntarily or introduced by the interviewer

following the interview guide. All female interviewees stated that they have personally experienced gender-based discrimination and harassment in the past.

"I had a very strange experience of hmm. . .harassment during my internship. Sexual harassment can happen to men, but I think it happened to me because I am a woman."(I5—Junior female attendee)

All man, woman, and gender minority interviewees (in total 7 of them) confirmed that they have witnessed or heard about such a phenomenon happening to their fellow female workers. This consensus contrasts with interviewees' varying degree of awareness regarding the overall topic of gender inequalities discussed in the previous section.

"A senior colleague told me that during her PhD, her PhD advisor said that since we now have female researchers, their jobs as researchers will be disrespected." (I1—Senior male attendee)

Although gender-based discrimination was found to be the most common type of discrimination against female professionals, 3 interviewees pointed out that sexual harassment also occurred often and incidents were rarely reported or dealt with at an organizational level. It is worth mentioning that the female interviewee of foreign nationality indicated her personal experience of intersectional discrimination, which underlines the particularity of discrimination due to her identities of being a woman as well as being a foreigner.

## Discussion

By implementing a quantitative and qualitative observation study on the JOBIM 2021 conference, we established and reproduced several key findings on how gender and sexual identity impact working conditions in STEM and more specifically at scientific conferences. Several indicators of equal gender representation improved significantly between the first edition of JOBIM (2000) and JOBIM 2021. Notably, parity was observed amongst attendees, keynote speakers, the program committee, chairmen and chairwomen of JOBIM 2021. For the first time, we reported the fraction of attendees identifying as LGBTQIA+: 9.5%. When accounting for attendees' demographics, women, gender minorities and junior academics were still strongly underrepresented in question askers. The qualitative analysis of attendees interviews provides potential explanation mechanisms to this picture.

The virtual setting of JOBIM 2021 was an opportunity to observe if the under-representation of women would persist in different conditions than the traditional live conferences. Interestingly the online format seemed to boost the registration of women, with the conference reaching parity in attendees for the first time in 2021. This increase in women registration might be due to the lesser logistic burden of attending virtual conferences enabling them to navigate, for instance, more easily child care and conference attendance. Indeed, parenthood still impacts female academics more than their male peers [33, 34].

Several factors could have mitigated the lesser number of questions from women at virtual conferences: question asking in written form and the possibility to ask anonymously. Yet, the underrepresentation of women in askers of written questions was similar to previous reports on live conferences [13, 14] and the anonymous questions remained marginal (10 questions overall). The virtual setting enabled us to identify askers more precisely and to count the total number of questions throughout the conference by asker. Interestingly, the vast majority (87%) of attendees did not ask any questions. Attendees asking numerous questions (3 or more throughout the conference) were men at 85.6%. Anecdotally, several observers stated that they were able to recall the name of the "serial askers" at the end of the conference, hinting that recurrent question asking is an efficient way to gain visibility in a field. To clarify if the women underrepresentation was merely a consequence of the leaky pipeline (attrition of women as we

climb the hierarchical ladder), we modeled the effect of gender, age, and professional status on the rate of question-asking. Accounting for seniority, gender still impacted the rate of question asking with women and gender minorities asking 50% less questions. Although the impact of seniority might be more understandable than the impact of gender as experience might help to formulate relevant questions, its extent was striking (4.6 times more questions for a senior academic). A pedagogical opportunity might be lost in scientific conferences if junior academics are not comfortable enough to ask questions. Interestingly, encouraging junior academics to ask the first question has been shown to mitigate women under-representation as well [13].

The fraction of members of the LGBTQIA+ community was comparable to the one reported in the French general population (9.5% compared to 8%) which contrast with previous reports stating that LGBTQIA+ community is under-represented in STEM [20, 23, 35]. However, this fraction might be overoptimistic since young attendees seem to be more willing to identify as LGBT and the majority of attendees at JOBIM are under 35. The estimate of the LGBT fraction in the French general population is also subject to controversy and is not, to our knowledge, reported in the scientific literature. The estimate provided here originates from a poll survey (Le regard des français sur l'homosexualité et la place des lgbt dans la société (2019)) conducted by the IFOP (Institut français d'opinion publique). In line with a previous report, more women identify as LGBT in our sample suggesting that gay men are more discriminated against in STEM than lesbian women [23]. While remaining relatively rare to this day with 13 attendees identifying outside of binary categories, the report of diverse gender identities stresses the value of proposing a variety of options other than male and female when collecting gender-related information. Concerning question asking, we report a tendency of LGBTQIA+ to be underrepresented. The underrepresentation of LGBTQIA+ in question asking is coherent with reports underlining that this community is marginalized in STEM [36]. Yet, this effect was no longer significant when accounting for gender and seniority. Due to the scarcity of data (5 questions asked by LGBTQIA+ members), it is unclear if the underrepresentation of LGBT in askers is coufounded by gender and seniority or if it has an effect on its own.

The post survey demonstrated that all genders prefer the same setting (live conferences) and are encouraged to ask questions by the same factors. While attendees noticed accurately the importance of seniority in question asking behavior, they were unaware of the effect of gender indicating the need for communication on this topic. Women and gender minorities reported that they endured more discrimination during conferences and were less motivated to ask questions. Potential underlying factors to these observations were provided by the qualitative analysis of attendee interviews. Among the recurrent themes were "Gender-based negative experience" and "Women are subjected to discrimination & harassment" highlighting the overall more negative experience of women and gender minorities in STEM as reported before [3, 25, 37]. Discouraging reactions to women speech were cited as causes of an hindered motivation to speak. Concerning sexual harassment, incident reported in interviews occurred exclusively to junior women. Although our sample is limited in size, this finding suggests that the power imbalance between junior female academics and senior male academics with a permanent position can be an enabling environment for harassment, which might explain why sexual harassment is higher in academia than in other working environments [37]. Female interviewees and gender minority noticed that their confidence was impacted by gender and in turn impacted ability to speak up. Hence, self-confidence might be the mediator between an unwelcoming environment and lesser ability to speak or self-promote rather than the primal cause. While this effect was not apparent in the post survey, women and gender minorities exhibit a higher degree of information on gender bias and more willingness to take action to correct it in interviews. This is coherent with previous reports stating that men are skeptical about the gender gap in general [14, 30, 31].

We believe that ensuring inclusivity should start already at the stage of research design and implementation; as researchers, we share the responsibility of acknowledging our own biases when conducting studies on such a complex topic. Although question asking is an interesting behavior to observe (public, not externally constrained) it pertains limitations. Indeed only 13% of attendees ask questions, hence the initial sample size of all attendees is shrink down to 101 askers. This limited sample might explain why we could not confirm the underrepresentation of LBGT attendees in question askers. While we present an original first attempt to combine qualitative and quantitative approaches, the number of interviews we were able to conduct was limited (7). Attendees accepting to go through an interview might not be entirely representative of all attendees. Notably, we were not able to interview a senior male attendee due to the lack of positive answers to our solicitations of members of this group. We believe that encouraging the systematic collection of data on gender but also on other axes (LGBTQIA+ belonging, racial, socio-economic) by institutions will help to address these limitations.

Our study documents the persistence of barriers against women and gender minorities in academia and provides hints on the most important problems to address and how to include them. We provide guidelines on the project web page for conference organizers as a first step to improve the current situation. We indeed think that large scientific conferences, by their collective nature, are the ideal opportunity to address the phenomena of gender-based inequity as well as to initiate changes in the STEM field.

## Supporting information

**S1 Fig. Overview of the study with the list of data collected, the main data processing steps and the main analysis steps.**
(TIF)

**S2 Fig. Predicted mean rate for junior and senior academics conditioned by their gender.**
By senior, we refer to an attendee older than 35 and with a permanent position. By junior, we refer to an attendee younger than 35 and with a short term contract.
(TIF)

**S3 Fig. Effect of the length of the question session.** A) Histogram of the number of questions asked at the end of the talk by type of sessions (color coded), B) Total number of questions asked by gender (indicated by the color of the bars) with respect to the total number of questions asked at the end of the talk (used as a proxy for the duration of question session).
(TIF)

**S4 Fig. Qualitative analysis of interview transcripts.** Each line of the transcript is assigned to a child code either derived from the research questions or created to represent the recurring topics. When all the transcripts have been processed, the resulting child codes are gathered into broader categories : parent codes. Finally parent codes are summarized in themes.
(TIF)

**S5 Fig.**
(PNG)

**S1 File. Observation guidelines and form.**
(PDF)

**S2 File. Interview consent form.**
(PDF)

**S3 File. Interview guide.**
(PDF)

**S1 Table. Confusion matrix between gender identified from first name versus self identified gender.** Each case is the count of attendees belonging to this category.
(XLSX)

**S2 Table. AIC by variables included in the Poisson regression.**
(XLSX)

**S3 Table. Initial codebook.**
(XLSX)

**S4 Table. Final codebook.**
(XLSX)

**S5 Table. Number of posters by gender and poster categories.**
(XLSX)

## Acknowledgments

We thank the SFBI and, in particular Julien Fumey, for providing data on previous JOBIM editions and collecting data on our behalf for this edition. We would like to thank the Institut Pasteur hub of bioinformatics and Biostastics for supporting this study. We thank Gillian Sandstrom for our fruitful scientific exchanges. We thank the JOBIM organizing committee for authorizing this study and enabling its presentation to the audience during the opening and closing sessions. We thank the team of observers and their invaluable contribution to this study : Elise Jacquemet, Sophie Schbath, Claudia Chica, Nicolas Maillet, Pascal Campagne. We thank Christophe Boetto and Stephan Fischer for their careful proofreading of the manuscript. We thank the reviewer for their thorough review and constructive feedback. Finally, we warmly thank all JOBIM attendees who entrusted us with their personal data, survey respondents and interviewees.

## Author Contributions

**Conceptualization:** Junhanlu Zhang, Rachel Torchet, Hanna Julienne.

**Data curation:** Junhanlu Zhang, Hanna Julienne.

**Formal analysis:** Junhanlu Zhang, Hanna Julienne.

**Investigation:** Junhanlu Zhang.

**Methodology:** Junhanlu Zhang, Hanna Julienne.

**Project administration:** Rachel Torchet.

**Resources:** Junhanlu Zhang, Rachel Torchet.

**Software:** Hanna Julienne.

**Supervision:** Hanna Julienne.

**Visualization:** Junhanlu Zhang, Rachel Torchet, Hanna Julienne.

**Writing – original draft:** Junhanlu Zhang, Hanna Julienne.

**Writing – review & editing:** Hanna Julienne.

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
