## [Decision Letter · Decision Letter 0]

16 Mar 2023

PONE-D-22-28787Gender-based disparities and biases in science: an observational study of a virtual conferencePLOS ONE

Dear Dr. Julienne,

Thank you for submitting your manuscript to PLOS ONE. After careful consideration, we feel that it has merit but does not fully meet PLOS ONE’s publication criteria as it currently stands. Therefore, we invite you to submit a revised version of the manuscript that addresses the points raised during the review process.

We look forward to receiving your revised manuscript.

Kind regards,

Andrew R. Dalby, PhD

Academic Editor

PLOS ONE

Journal Requirements:

4. Please upload a new copy of All Figures as the detail is not clear. Please follow the link for more information: " ext-link-type="uri" xlink:type="simple">https://blogs.plos.org/plos/2019/06/looking-good-tips-for-creating-your-plos-figures-graphics/"
https://blogs.plos.org/plos/2019/06/looking-good-tips-for-creating-your-plos-figures-graphics/

Reviewers' comments:

Reviewer's Responses to Questions

**Comments to the Author**

1. Is the manuscript technically sound, and do the data support the conclusions?

Reviewer #1: Yes

2. Has the statistical analysis been performed appropriately and rigorously? 

Reviewer #1: Yes

3. Have the authors made all data underlying the findings in their manuscript fully available?

Reviewer #1: Yes

4. Is the manuscript presented in an intelligible fashion and written in standard English?

Reviewer #1: Yes

5. Review Comments to the Author

Reviewer #1: This is an important study with useful recommendations.

Authors give quantitative data on an Online/virtual conference with qualitative individual interviews and address gender and sexual orientation. The main weakness is in the numbers of individuals interviewed, and this should be clarified in the data section itself. Still, this is a useful report.

At the top of page 7, authors state:

“women are subjected to gender-based discrimination and harassment during scientific activities, including academic conferences.”

It would be more accurate to state

“women report experiencing gender-based discrimination and harassment during scientific activities, including academic conferences.”

Believing one has experienced discrimination is not the same thing as discrimination. Perceptions may not be reality.

On page 7 what is meant by ‘surrounding work atmosphere.” Can you give some details? examples?

Similarly, how was she “discouraged to become a scientific researcher.” What was said? By whom?

Authors state. On p. “Furthermore, the findings also show that women and gender minority tend to improve their level of confidence through self-acceptance”. What evidence is there for this statement?

At the top of Page 9 authors wrote “Gender-based discrimination and harassment in the workplace6, including academic conferences, were one of the most frequently discussed phenomena during the interviews.” Is that because the interviewers specifically asked about this? Or did interviewees bring it up on their own?

Page 9 authors wrote “All man, woman, and gender minority interviewees confirmed that they have witnessed or heard about such a phenomenon happening to their fellow female workers.” Please add the exact number after the word interviewees, how many people was ‘all’? 6?

Page 10, the numbers here are too low to support this statement

“The fraction of members of the LGBTQIA+ community was higher than in the French general population (9.5% compared to 8%)”. You cannot tell any real difference between 9.5 and 8% with numbers this low.

Minor comments/corrections:

Near the bottom of page 2, Authors have accidentally used the wrong wording here

“We attempt to step out of the non-binary vision of gender by”

From the context, the authors meant to write

“We attempt to step out of the historical or typical binary vision of gender by”

In the intro, when authors describe the “underrepresentation of minorities” do they mean racial/ethnic minorities or are they including women in ‘minorities’ since women are minorities in science. Clarify.

Near the top of page 4, authors state

“Yet, amongst

the 13 JOBIM editions with available data, 6 significantly lacked women contributed speakers.”

I am uncertain what the phrase ‘women contributed speakers’ means. Does this mean that some people are chosen to give oral presentations while others are not? If so the phrasing might be

“Yet, amongst

the 13 JOBIM editions with available data, 6 significantly lacked women who were invited to be speakers.”

On page 4, the authors wrote

“This overall effect was modulated by session type and was

especially stark in 'Mini-symposia' and less obvious in the contributed talks (see Fig. 2A).”

Do the authors mean that mini symposia are more prestigious or have larger audiences than contributed talks? I’m guessing that is what is meant, but I am unsure. Please clarify.

On page 5, I believe the authors mean to write

“Even though JOBIM is considered as

student-friendly conference, the effect of age and professional status weighed heavily on the rate of questions

asked with senior academics asking 4.6 times more questions than junior academics.”

On page 5, authors wrote

“The results could differ from the precedent section due to a few questions asked orally by the attendees, the

questions asked by the chairman, and the selection effect of the chairman when reading questions out loud

from the chatbox.”

I think authors may mean the ‘preceding’ section rather than precedent if they mean the section prior to this one.

Second, the authors use the word ‘chairman’. In the US, we often just say the “chair’ with the same meaning, except it does not mean that the person is a man. In the same paragraph, the authors use chairperson, so I am now confused by what they mean by chairman.

I do not understand the meaning at the bottom of page 5 “we did not find a significant

positive effect of the length of question session on women and gender minority representation (see Fig. S3B).”

page 6 please clarify this “Hence, our results provide a slightly different information: the proportion

of respondents thinking that the factor matters.” Explain exactly what you mean.

At the top of page 7, it should state

“women, sexual and gender minority attendees report negative experiences

based on sexual orientation, gender identity and expression in professional contexts”

Similarly the heading on page 7 should read

Women and gender minority attendees report negative experiences based on sexual orien-

tations, gender identities, and expressions

Page 11.

This is coherent consistent with previous reports stating that men need are skeptical about the gender gap in general[14, 27, 28].

There is a typo on the website under “support minorities in question asking”. The word ‘talking’ is present, when it should be ‘taking”

6. PLOS authors have the option to publish the peer review history of their article (what does this mean?). If published, this will include your full peer review and any attached files.

Reviewer #1: No

---

## [Author Response · Author response to Decision Letter 0]

22 May 2023

Dear Reviewer,

Please find below our point by point answer to your concerns.

With kind regards,

Hanna Julienne, Ph.D.

Statistical Genetics Group,

Institut Pasteur 

hanna.julienne@pasteur.fr

5. Review Comments to the Author

Reviewer #1: This is an important study with useful recommendations.

Authors give quantitative data on an Online/virtual conference with qualitative individual interviews and address gender and sexual orientation. The main weakness is in the numbers of individuals interviewed, and this should be clarified in the data section itself. Still, this is a useful report.

We thank the reviewer for their appreciation of the importance of our work, their thorough review, and pointing out the main limitation of our study. 

Concerning the number of interviews conducted, we clarified the scope of the qualitative section by adding the following text in the Material and method “Semi-structured and in-depth interviews” paragraph. 

“The number of interviews we were able to conduct is relatively low (7) and our qualitative observations might not generalize to the complete academic population. Yet, these interviews provide valuable examples of career trajectories impacted by gender and by the LGBTQIA+ status. This contextualization can help generate hypotheses on why women and gender minorities ask less questions than men.”

We equally repeated this clarification at the beginning of the result section discussing qualitative findings:

“To contextualize quantitative findings we conducted in depth interviews on a representative set of participants. The number of interviews we were able to conduct is relatively low (7) and our qualitative observations might not generalize to the complete academic population. Yet, these interviews provide valuable examples on how gender and LGBTQIA+ status can impact career trajectories which could lead to a lesser comfort in question asking.”

Throughout the Result section describing qualitative finding we removed intensity adverbs (e.g. strongly, significantly) and modified strong affirmative wording (e.g. change “is associated” to “might be associated”) to avoid any overstatements of our results.

At the top of page 7, authors state:

“women are subjected to gender-based discrimination and harassment during scientific activities, including academic conferences.”

It would be more accurate to state

“women report experiencing gender-based discrimination and harassment during scientific activities, including academic conferences.”

Believing one has experienced discrimination is not the same thing as discrimination. Perceptions may not be reality.

We thank the reviewer for the pertinent remark and suggestion. We updated the manuscript text accordingly.

On page 7 what is meant by ‘surrounding work atmosphere.” Can you give some details? examples?

“Surrounding work atmosphere” refers to the general professional environment of the interviewees. In the following text, one example is given by Interviewee I4 who reported witnessing in her work environment negative reactions received by a female team leader from other male team leaders when posing questions or presenting. We thank the reviewer for pointing out this unclear expression. To avoid potential confusion, we removed “surrounding work atmosphere”. In fact, we think such a modification will not affect the original meaning of the phrase.

Similarly, how was she “discouraged to become a scientific researcher.” What was said? By whom?

We added the supportive evidence by I3 to the text.

Authors state. On p. “Furthermore, the findings also show that women and gender minority tend to improve their level of confidence through self-acceptance”. What evidence is there for this statement?

We thank the reviewer for pointing out that our statement lacked thorough support in the previous version of the manuscript. We substantiated this point by adding supportive evidence formulated by I2 and I1 to the text. We also reformulated the statement to a more hypothetical form “Furthermore, the findings also show that women and gender minorities might improve their level of confidence through self-acceptance.”

We also contextualize this observation with studies on the evolution of self-esteem through life reporting women have lessened self-esteem compared to men:

“These observations correspond in part with studies reporting that women tend to have a lessened self-esteem compared to men, especially during early adulthood, and start recovering partially during their forties (Robins et al, 2002; Robins et al, 2005). Our observation suggests that the origin of this recovery could be internal rather than external. Confirming these hypotheses would need further investigations.”

At the top of Page 9 authors wrote “Gender-based discrimination and harassment in the workplace6, including academic conferences, were one of the most frequently discussed phenomena during the interviews.” Is that because the interviewers specifically asked about this? Or did interviewees bring it up on their own?

We thank the reviewer for raising this valuable question. To answer the question, both situations occurred. In some cases, the interviewer brought the topic up by following the interview guide. Specific examples are given as following:

For example, during the interview with I4, the interviewee commented on the phenomenon of gender discrimination in her current workplace without the subject being mentioned by the interviewer. I4 talked about gender discrimination following the interviewer’s general question about the impact of gender identity on her professional life. Similarly, I5 voluntarily talked about her experience of sexual harassment during the interview. However, it also happened that discussion on such a subject occurred during the interview because the interviewer prompted the topic of gender-based discrimination or harassment. As an example, the interviewer asked the question to I6: “Have you participated in or heard of discussions around gender-related topics? For example, gender-based stereotypes or discrimination.”

To clarify this point, we added the following sentence as a complementary information: “The subject of discussion was either brought up by the interviewees voluntarily or introduced by the interviewer following the interview guide.”

In addition to this point, what we would like to highlight here is the consensus among interviewees that gender-based discrimination and harassment are common phenomena in the workplace. All interviewees, despite the differences in their gender identity and seniority, all reported either hearing about or witnessing such events or experiencing them first hand in their workplace. To this end, we added the following sentence to the text: 

“All man, woman, and gender minority interviewees (in total 7 of them) confirmed that they have witnessed or heard about such a phenomenon happening to their fellow female workers. This consensus contrasts with interviewees' varying degree of awareness regarding the overall topic of gender inequalities discussed in the previous section.”

Page 9 authors wrote “All man, woman, and gender minority interviewees confirmed that they have witnessed or heard about such a phenomenon happening to their fellow female workers.” Please add the exact number after the word interviewees, how many people was ‘all’? 6?

We thank the reviewer for the pertinent remark and suggestion. We added the information to the text. In total there were 7 interviewees.

Page 10, the numbers here are too low to support this statement

“The fraction of members of the LGBTQIA+ community was higher than in the French general population (9.5% compared to 8%)”. You cannot tell any real difference between 9.5 and 8% with numbers this low.

We thank the reviewer for noticing this poor wording on our part. Indeed, 8% is included in the confidence interval of our point estimate (9.5%). Hence, the fraction of LGBTQIA+ participant in the JOBIM conference seems comparable to the general French population. We modified the text as follows:

“The fraction of members of the LGBTQIA+ community was comparable to the one reported in the French general population (9.5% compared to 8%), which contrast with previous reports stating that LGBTQIA+ community is under-represented in STEM [20, 22, 34]”

We believe that the rest of the sentence, which contextualizes this percentage against previous reports, remains pertinent.

Minor comments/corrections:

We thank the reviewer for their careful proofreading of the manuscript. We inserted all proposed suggestions.

Near the bottom of page 2, Authors have accidentally used the wrong wording here

“We attempt to step out of the non-binary vision of gender by”

From the context, the authors meant to write

“We attempt to step out of the historical or typical binary vision of gender by”

We thank the reviewer for noticing this error. We inserted the proposed suggestion in our manuscript.

In the intro, when authors describe the “underrepresentation of minorities” do they mean racial/ethnic minorities or are they including women in ‘minorities’ since women are minorities in science. Clarify.

We thank the reviewer for pointing out that minorities could be misinterpreted when referring to women. In sociology, women are frequently referred to as minority not necessarily in terms of number but rather because they have lessened access to power and opportunity than men. 

Nonetheless, our usage of minority in the introduction was vague (as it could refer to racial minorities) and more obscuring than enlightening. We replaced “minorities” by “women and gender minority” while keeping a footnote specifying that by gender minority we refer to people identifying outside of the binary categories of male and female. 

Near the top of page 4, authors state

“Yet, amongst

the 13 JOBIM editions with available data, 6 significantly lacked women contributed speakers.”

I am uncertain what the phrase ‘women contributed speakers’ means. Does this mean that some people are chosen to give oral presentations while others are not? If so the phrasing might be

“Yet, amongst

the 13 JOBIM editions with available data, 6 significantly lacked women who were invited to be speakers.”

Indeed, “contributed speakers” refers to a person who submitted an abstract to the conference program committee and were selected to give an oral presentation. We clarified this expression by inserting a footnote to define it.

On page 4, the authors wrote

“This overall effect was modulated by session type and was

especially stark in 'Mini-symposia' and less obvious in the contributed talks (see Fig. 2A).”

Do the authors mean that mini symposia are more prestigious or have larger audiences than contributed talks? I’m guessing that is what is meant, but I am unsure. Please clarify.

In the JOBIM conference, mini symposia are a series of technical talks organized around a theme usually featuring mid-career researchers. While it might be of importance, the reason for a stronger underrepresentation of women in question askers in this type of talks is not completely clear at this stage. We modified the text to provide the element of context necessary to understand what “mini symposia” refers to while avoiding overstating the observed tendency.

“We noticed a tendency for a stronger underrepresentation in 'Mini-symposia', which are series of technical talks featuring invited speakers, and less obvious in the contributed talks, which usually feature more junior speakers (Fig. 2A).”

On page 5, I believe the authors mean to write

“Even though JOBIM is considered as

student-friendly conference, the effect of age and professional status weighed heavily on the rate of questions

asked with senior academics asking 4.6 times more questions than junior academics.”

On page 5, authors wrote

“The results could differ from the precedent section due to a few questions asked orally by the attendees, the

questions asked by the chairman, and the selection effect of the chairman when reading questions out loud

from the chatbox.”

I think authors may mean the ‘preceding’ section rather than precedent if they mean the section prior to this one.

Second, the authors use the word ‘chairman’. In the US, we often just say the “chair’ with the same meaning, except it does not mean that the person is a man. In the same paragraph, the authors use chairperson, so I am now confused by what they mean by chairman.

The above suggestions have been inserted.

I do not understand the meaning at the bottom of page 5 “we did not find a significant

positive effect of the length of question session on women and gender minority representation (see Fig. S3B).”

The wording has been clarified to: “we did not find a significant positive effect of the total number of questions asked during the question session on women and gender minority representation”.

page 6 please clarify this “Hence, our results provide a slightly different information: the proportion

of respondents thinking that the factor matters.” Explain exactly what you mean.

We clarified this paragraph by providing a brief description of Likert scale. We removed the sentence point out by the reviewer as we thought it was indeed unclear:

“This might be due to the survey design: used Likert scales (where respondents are asked how much a factor matters using a scale from one to five) whereas we asked respondents to say if a factor mattered or not in a binary way.”

At the top of page 7, it should state

“women, sexual and gender minority attendees report negative experiences

based on sexual orientation, gender identity and expression in professional contexts”

Similarly the heading on page 7 should read

Women and gender minority attendees report negative experiences based on sexual orientations, gender identities, and expressions

We thank the reviewer for pointing out our lack of accuracy in describing the concluding statements. We modified both sentences according to the reviewer’s suggestions.

Page 11.

This is coherent consistent with previous reports stating that men need are skeptical about the gender gap in general[14, 27, 28].

There is a typo on the website under “support minorities in question asking”. The word ‘talking’ is present, when it should be ‘taking”

We thank the reviewer for their careful proofreading. We inserted the proposed suggestions.

---

## [Editor Report · Decision Letter 1]

24 May 2023

Gender-based disparities and biases in science: an observational study of a virtual conference

PONE-D-22-28787R1

Dear Dr. Julienne,

We’re pleased to inform you that your manuscript has been judged scientifically suitable for publication and will be formally accepted for publication once it meets all outstanding technical requirements.

Kind regards,

Andrew R. Dalby, PhD

Academic Editor

PLOS ONE
---

## [Editor Report · Acceptance letter]

30 May 2023

PONE-D-22-28787R1 

Gender-based disparities and biases in science: an observational study of a virtual conference 

Dear Dr. Julienne:

I'm pleased to inform you that your manuscript has been deemed suitable for publication in PLOS ONE. Congratulations! Your manuscript is now with our production department. 

Kind regards, 

on behalf of

Dr. Andrew R. Dalby 

Academic Editor

PLOS ONE